# A Multi-Resolution Framework for U-Nets with Applications to Hierarchical VAEs

**Fabian Falck** [*,1,3,4]   **Christopher Williams** [*,1]   **Dominic Danks** [2,4]   **George Deligiannidis** [1]
**Christopher Yau** [1,3,4]   **Chris Holmes** [1,3,4]   **Arnaud Doucet** [1]   **Matthew Willetts** [4]
[1]University of Oxford  [2]University of Birmingham
[3]Health Data Research UK  [4]The Alan Turing Institute
{fabian.falck, williams, deligian, cholmes, doucet}@stats.ox.ac.uk,
{ddanks, cyau, mwilletts}@turing.ac.uk

## Abstract

U-Net architectures are ubiquitous in state-of-the-art deep learning, however their regularisation properties and relationship to wavelets are understudied. In this paper, we formulate a multi-resolution framework which identifies U-Nets as finite-dimensional truncations of models on an infinite-dimensional function space. We provide theoretical results which prove that average pooling corresponds to projection within the space of square-integrable functions and show that U-Nets with average pooling implicitly learn a Haar wavelet basis representation of the data. We then leverage our framework to identify state-of-the-art hierarchical VAEs (HVAEs), which have a U-Net architecture, as a type of two-step forward Euler discretisation of multi-resolution diffusion processes which flow from a point mass, introducing sampling instabilities. We also demonstrate that HVAEs learn a representation of time which allows for improved parameter efficiency through weight-sharing. We use this observation to achieve state-of-the-art HVAE performance with half the number of parameters of existing models, exploiting the properties of our continuous-time formulation.

## 1  Introduction

U-Net architectures are extensively utilised in modern deep learning models. First developed for image segmentation in biomedical applications [1], U-Nets have been widely applied for text-to-image models [2], image-to-image translation [3], image restoration [4, 5], super-resolution [6], and multiview learning [7], amongst other tasks [8]. They also form a core building block as the neural architecture of choice in state-of-the-art generative models, particularly for images, such as HVAEs [9, 10, 11, 12] and diffusion models [2, 13, 14, 15, 16, 17, 18, 19, 20]. In spite of their empirical success, it is poorly understood why U-Nets work so well, and what regularisation they impose.

In likelihood-based generative modelling, various model classes are competing for superiority, including normalizing flows [21, 22], autoregressive models [23, 24], diffusion models, and hierarchical variational autoencoders (HVAEs), the latter two of which we focus on in this work. HVAEs form groups of latent variables with a conditional dependence structure, use a U-Net neural architecture, and are trained with the typical VAE ELBO objective (for a detailed introduction to HVAEs, see Appendix B). HVAEs show impressive synthesis results on facial images, and yield competitive likelihood performance, consistently outperforming the previously state-of-the-art autoregressive models, VAEs and flow models on computer vision benchmarks [9, 10]. HVAEs have undergone a journey

---

[*]Equal contribution.

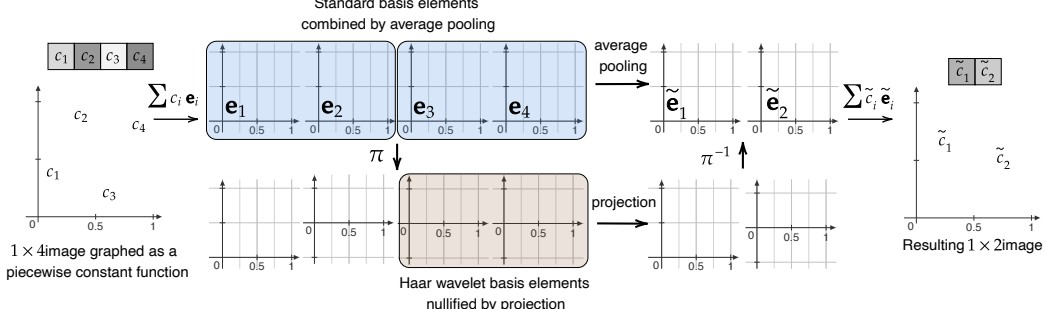

Figure 1: U-Nets with average pooling learn a Haar wavelet basis representation of the data.

of design iterations and architectural improvements in recent years, for example the introduction a deterministic backbone [25, 26, 27] and ResNet elements [28, 29] with shared parameters between the inference and generative model parts. There has also been a massive increase in the number of latent variables and overall stochastic depth, as well as the use of different types of residual cells in the decoder [9, 10] (see §4 and Fig. A.1 for a detailed discussion). However, a theoretical understanding of these choices is lacking. For instance, it has not been shown why a residual backbone may be beneficial, or what the specific cell structures in VDVAE [9] and NVAE [10] correspond to, or how they could be improved.

In this paper we provide a theoretical framework for understanding the latent spaces in U-Nets, and apply this to HVAEs specifically. Doing so allows us to relate HVAEs to diffusion processes, and also to motivate a new type of piecewise time-homogenenous model which demonstrates state-of-the-art performance with approximately half the number of parameters of a VDVAE [9]. More formally, our contributions are as follows: **(a)** We provide a multi-resolution framework for U-Nets. We formally define U-Nets as as acting over a multi-resolution hierarchy of $L^2([0,1]^2)$. We prove that average pooling is a conjugate operation to projection in the Haar wavelet basis within $L^2([0,1]^2)$. We use this insight to show how U-Nets with average pooling implicitly learn a Haar wavelet basis representation of the data (see Fig. 1), helping to characterise the regularisation within U-Nets. **(b)** We apply this framework to state-of-the-art HVAEs as an example, identifying their residual cell structure as a type of two-step forward Euler discretisation of a multi-resolution diffusion bridge. We uncover that this diffusion process flows from a point mass, which causes instabilities, for instance during sampling, and identify parameter redundancies through our continuous-time formulation. Our framework both allows us to understand the heuristic choices of existing work in HVAEs and enables future work to optimise their design, for instance their residual cell. **(c)** In our experiments, we demonstrate these sampling instabilities and train HVAEs with the largest stochastic depth ever, achieving state-of-the-art performance with half the number of parameters by exploiting our theoretical insights. We explain these results by uncovering that HVAEs secretly represent time in their state and show that they use this information during training. We finally provide extensive ablation studies which, for instance, rule out other potential factors which correlate with stochastic depth, show the empirical gain of multiple resolutions, and find that Fourier features (which discrete-time diffusion models strongly benefit from [19]) do not improve performance in the HVAE setting.

## 2 The Multi-Resolution Framework

A grayscale image with infinite resolution can be thought of as the graph[2] of a two-dimensional function over the unit square. To store these infinitely-detailed images in computers, we project them to some finite resolution. These projections can still be thought of as the graphs of functions with support over the unit square, but they are piecewise constant on finitely many intervals or 'pixels', e.g. $512^2$ pixels, and we store the function values obtained at these pixels in an array or 'grid'. The relationship between the finite-dimensional version and its infinitely-fine counterpart depends entirely on how we construct this projection to preserve the details we wish to keep. One approach is to prioritise preserving the large-scale details of our images, so unless closely inspected, the projection

---

[2]For a function $f(\cdot)$, its graph is the set $\bigcup_{x \in [0,1]^2} \{x, f(x)\}$.

is indistinguishable from the original. This can be achieved with a multi-resolution projection [30] of the image. In this section we introduce a *multi-resolution framework* for constructing neural network architectures that utilise such projections, prove what regularisation properties they impose, and show as an example how HVAEs with a U-Net [1] architecture can be interpreted in our framework. Proofs of all theorems in the form of an extended exposition of our framework can be found in Appendix A.

## 2.1 Multi-Resolution Framework: Definitions and Intuition

What makes a multi-resolution projection good at prioritising large-scale details can be informally explained through the following thought experiment. Imagine we have an image, represented as the graph of a function, and its finite-dimensional projection drawn on the wall. We look at the wall, start walking away from it and stop when the image and its projection are indistinguishable by eye. The number of steps we took away from the wall can be considered our measure of 'how far away' the approximation is from the underlying function. The goal of the multi-resolution projection is therefore to have to take as few steps away as possible. The reader is encouraged to physically conduct this experiment with the examples provided in Appendix B.1. We can formalise the aforementioned intuitions by defining a *multi-resolution hierarchy* [30] of sub-spaces we may project to:

**Definition 1.** [*Daubechies (1992)* [30]] Given a nested sequence of *approximation spaces* $\cdots \subset V_1 \subset V_0 \subset V_{-1} \subset \cdots$, $\{V_{-j}\}_{j \in \mathbb{Z}}$ is a *multi-resolution hierarchy* of the function space $L^2(\mathbb{R}^m)$ if: (**A1**) $\overline{\bigcup_{j \in \mathbb{Z}} V_{-j}} = L^2(\mathbb{R}^m)$; (**A2**) $\bigcap_{j \in \mathbb{Z}} V_{-j} = \{0\}$; (**A3**) $f(\cdot) \in V_{-j} \Leftrightarrow f(2^j \cdot) \in V_0$; (**A4**) $f(\cdot) \in V_0 \Leftrightarrow f(\cdot - n) \in V_0$ for $n \in \mathbb{Z}$. For a compact set $\mathbb{X} \subset \mathbb{R}^m$, a *multi-resolution hierarchy* of $L^2(\mathbb{X})$ is $\{V_{-j}\}_{j \in \mathbb{Z}}$ as defined above, restricting functions in $V_{-j}$ to be supported on $\mathbb{X}$.

In Definition 1, the index $j$ references how many steps we took in our thought experiment, so negative $j$ corresponds to 'zooming in' on the images. The original image[3] is a member of $L^2([0, 1]^2)$, the space of square-integrable functions on the unit square, and its finite projection to $2^j \cdot 2^j$ many pixels is a member of $V_{-j}$. Images can be represented as piecewise continuous functions in the subspaces $V_{-j} = \{f \in L^2([0, 1]) \mid f|_{[2^{-j} \cdot k, 2^{-j} \cdot (k+1))} = c_k, k \in \{0, \ldots, 2^j - 1\}, c_k \in \mathbb{R}\}$. The nesting property $V_{-j+1} \subset V_{-j}$ ensures that any image with $(2^{j-1})^2$ pixels can also be represented by $(2^j)^2$ pixels, but at a higher resolution. Assumption (**A1**) states that with infinitely many pixels, we can describe any infinitely detailed image. In contrast, (**A2**) says that with no pixels, we cannot approximate any images. Assumptions (**A3**) and (**A4**) allow us to form a basis for images in any $V_{-j}$ if we know the basis of $V_0$. One basis made by extrapolating from $V_0$ in this way is known as a *wavelet basis* [30]. Wavelets have proven useful for representing images, for instance in the JPEG standard [31], and are constructed to be orthonormal.

Now suppose we have a probability measure $\nu_\infty$ over infinitely detailed images represented in $L^2([0, 1]^2)$ and wish to represent it at a lower resolution. Similar to how we did for infinitely detailed images, we want to project the measure $\nu_\infty$ to a lower dimensional measure $\nu_j$ on the finite dimensional space $V_{-j}$. In extension to this, we want the ability to reverse this projection so that we may sample from the lower dimensional measure and create a generative model for $\nu_\infty$. We would like to again prioritise the presence of large-scale features of the original image within the lower dimensional samples. We do this by constructing a *multi-resolution bridge* from $\nu_\infty$ to $\nu_j$, as defined below.

**Definition 2.** Let $\mathbb{X} \subset \mathbb{R}^m$ be compact, $\{V_{-j}\}_{j=0}^\infty$ be a multi-resolution hierarchy of scaled so $L^2(\mathbb{X}) = \overline{\bigcup_{j \in \mathbb{N}_0} V_{-j}}$ and $V_0 = \{0\}$. If $\mathbb{D}(L^2(\mathbb{X}))$ is the space of probability measures over $L^2(\mathbb{X})$, then a family of probability measures $\{\nu_t\}_{t \in [0,1]}$ on $L^2(\mathbb{X})$ is a *multi-resolution bridge* if:

(i) there exist increasing times $\mathcal{I} := \{t_j\}_{j \in \mathbb{N}_0}$ where $t_0 = 0$, $\lim_{j \to \infty} t_j = 1$, such that $s \in [t_j, t_{j+1})$ implies $\text{supp}(\nu_s) \subset V_{-j}$, i.e $\nu_s \in \mathbb{D}(V_{-j})$; and,

(ii) for $s \in (0, 1)$, the mapping $s \mapsto \nu_s$ is continuous for $s \in (t_j, t_{j+1})$ for some $j$.

The continuous time dependence in Definition 2 plays a movie of the measure $\nu_0$ supported on $V_0$ growing to $\nu_\infty$, a measure on images with infinite resolution. At a time interval $[t_j, t_{j+1})$, the space $V_{-j}$ which the measure is supported on is fixed. We may therefore define a finite-dimensional model

---

[3]We here focus on grayscale, squared images for simplicity, but note that our framework can be seamlessly extended to colour images with a Cartesian product $L^2([0, 1]^2) \times L^2([0, 1]^2) \times L^2([0, 1]^2)$, and other continuous signals such as time series.

transporting probability measures within $V_{-j}$, but at $t_{j+1}$ the support flows over to $V_{-j-1}$. Given a multi-resolution hierarchy, we may glue these finite models, each acting on a disjoint time interval, together in a unified fashion. In Theorem 1 we show this for the example of a continuous-time multi-resolution diffusion process truncated up until some time $t_J = T \in (0,1)$ and in the *standard basis* discussed in §2.2, which will be useful when viewing HVAEs as discretisations of diffusion processes on functions in §2.3.

**Theorem 1.** *Let $B_j : [t_j, t_{j+1}] \times \mathbb{D}(V_{-j}) \mapsto \mathbb{D}(V_{-j})$ be a linear operator (such as a diffusion transition kernel, see Appendix A) for $j < J$ with coefficients $\mu^{(j)}, \sigma^{(j)} : [t_j, t_{j+1}] \times V_{-j} \mapsto V_{-j}$, and define the natural extensions within $V_{-J}$ in bold, i.e. $\boldsymbol{B}_j := B_j \oplus \boldsymbol{I}_{V_{-j}^\perp}$. Then the operator $\boldsymbol{B} : [0,T] \times \mathbb{D}(V_{-J}) \mapsto \mathbb{D}(V_{-J})$ and the coefficients $\boldsymbol{\mu}, \boldsymbol{\sigma} : [0,T] \times V_{-J} \mapsto V_{-J}$ given by*

$$\boldsymbol{B} := \sum_{j=0}^{J} \mathbb{1}_{[t_j, t_{j+1})} \cdot \boldsymbol{B}_j, \quad \boldsymbol{\mu} := \sum_{j=0}^{J} \mathbb{1}_{[t_j, t_{j+1})} \cdot \boldsymbol{\mu}^{(j)}, \quad \boldsymbol{\sigma} := \sum_{j=0}^{J} \mathbb{1}_{[t_j, t_{j+1})} \cdot \boldsymbol{\sigma}^{(j)},$$

*induce a multi-resolution bridge of measures from the dynamics for $t \in [0,T]$ and on the standard basis as $dZ_t = \boldsymbol{\mu}_t(Z_t)dt + \boldsymbol{\sigma}_t(Z_t)dW_t$ (see Appendix A.4 for details) for $Z_t \in V_{-j}$ for $t \in [t_j, t_{j+1})$, i.e. a multi-resolution diffusion process.*

The concept of a multi-resolution bridge will become important in Section 2.2 where we will show that current U-Net bottleneck structures used for unconditional sampling impose a multi-resolution bridge on the modelled densities. To preface this, we here provide a description of a U-Net within our framework, illustrated in 2. Consider $B_{j,\theta}, F_{j,\theta} : \mathbb{D}(V_{-j}) \rightarrow \mathbb{D}(V_{-j})$ as the forwards and backwards passes of a U-Net on resolution $j$. Further, let $P_{-j+1} : \mathbb{D}(V_{-j}) \rightarrow \mathbb{D}(V_{-j+1})$ and $E_{-j} : \mathbb{D}(V_{-j+1}) \rightarrow \mathbb{D}(V_{-j})$ be the projection (here: average pooling) and embedding maps (e.g. interpolation), respectively. When using an $L^2$-reconstruction error, a U-Net [1] architecture implicitly learns a sequence of models $\mathbf{B}_{j,\phi} : \mathbb{D}(V_{-j+1}) \times \mathbb{D}(V_{-j+1}^\perp) \mapsto \mathbb{D}(V_{-j})$ due to the orthogonal decomposition $V_{-j} = V_{-j+1} \oplus U_{-j+1}$ where $U_{-j+1} := V_{-j} \cap V_{-j+1}^\perp$. The backwards operator for the U-Net has a (bottleneck) input from $\mathbb{D}(V_{-j+1})$ and a (skip) input yielding information from $\mathbb{D}(V_{-j+1}^\perp)$. A simple *bottleneck* map $U_{j,\theta} : \mathbb{D}(V_{-j}) \rightarrow \mathbb{D}(V_{-j})$ (without skip connection) is given by

Figure 2: A U-Net in our multi-resolution framework. See Appendix B.2 for details.

$$U_{j,\theta} := B_{j,\theta} \circ E_{-j} \circ P_{-j+1} \circ F_{j,\theta}, \quad (1)$$

and a U-Net bottleneck with skip connection is

$$\mathbf{U}_{j,\phi} := B_{j,\phi}(E_{-j} \circ P_{-j+1} \circ F_{j,\theta}, F_{j,\theta}). \quad (2)$$

In HVAEs, the map $\mathbf{U}_{j,\phi} : \mathbb{D}(V_{-j}) \rightarrow \mathbb{D}(V_{-j})$ is trained to be the identity by minimising reconstruction error, and further shall approximate $U_{j,\theta} \approx \mathbf{U}_{j,\phi}$ via a KL divergence. The $L^2$-reconstruction error for $\mathbf{U}_{j,\phi}$ has an orthogonal partition of the inputs from $V_{-j+1} \times V_{-j}$, hence the only new subspace added is $U_{-j+1}$. As each orthogonal $U_{-j+1}$ is added sequentially in HVAEs, the skip connections induce a multi-resolution structure of this hierarchical neural network structure. What we will investigate in Theorem 3 is the regularisation imposed on this partitioning by enforcing $U_{j,\theta} \approx \mathbf{U}_{j,\phi}$, as is often enforced for generative models with VAEs.

## 2.2 The regularisation property imposed by U-Net architectures with average pooling

Having defined U-Net architectures within our multi-resolution framework, we are now interested in the regularisation they impose. We do so by analysing a U-Net when skip connections are absent, so that we may better understand what information is transferred through each skip connection when they are present. In practice, a pixel representation of images is used when training U-Nets, which we henceforth call the *standard basis* (see A.2, Eq. (A.9)). The standard basis is not convenient to derive theoretical results. It is instead preferable to use a basis natural to the multi-resolution bridge imposed by a U-Net with a corresponding projection operation, which for average pooling is the *Haar (wavelet) basis* [32] (see Appendix A.2). The Haar basis, like a Fourier basis, is an orthonormal basis

of $L^2(\mathbb{X})$ which has desirable $L^2$-approximation properties. We formalise this in Theorem 2 which states that the dimension reduction operation of average pooling in the standard basis is a conjugate operation to co-ordinate projection within the Haar basis (details are provided in Appendix A.2).

**Theorem 2.** *Given $V_{-j}$ as in Definition 1, let $x \in V_{-j}$ be represented in the standard basis $\mathbf{E}_j$ and Haar basis $\mathbf{\Psi}_j$. Let $\pi_j : \mathbf{E}_j \mapsto \mathbf{\Psi}_j$ be the change of basis map illustrated in Fig. 3, then we have the conjugacy $\pi_{j-1} \circ pool_{-j,-j+1} = proj_{V_{-j+1}} \circ \pi_j$.*

Theorem 2 means that if we project an image from $V_{-j}$ to $V_{-j+1}$ in the Haar wavelet basis, we can alternatively view this as changing to the standard basis via $\pi_j^{-1}$, performing average pooling, and reverting back via $\pi_{j-1}$ (see Figure 3). This is important because the Haar basis is orthonormal, which in Theorem 3 allows us to precisely quantify what information is lost with average pooling.

$$
\begin{array}{ccc}
(V_{-j}, \mathbf{E}_j) & \xrightarrow{\ pool_{-j,-j+1}\ } & (V_{-j+1}, \mathbf{E}_{j-1}) \\
\uparrow{\scriptstyle \pi_j^{-1}} & & \downarrow{\scriptstyle \pi_{j-1}} \\
(V_{-j}, \mathbf{\Psi}_j) & \dashrightarrow[\ proj_{V_{-j+1}}\ ] & (V_{-j+1}, \mathbf{\Psi}_{j-1})
\end{array}
$$

Figure 3: The function space $V_{-j}$ remains the same, but the basis changes under $\pi_j$.

**Theorem 3.** *Let $\{V_{-j}\}_{j=0}^J$ be a multi-resolution hierarchy of $V_{-J}$ where $V_{-j} = V_{-j+1} \oplus U_{-j+1}$, and further, let $F_{j,\phi}, B_{j,\theta} : \mathbb{D}(V_{-j}) \mapsto \mathbb{D}(V_{-j})$ be such that $B_{j,\theta}F_{j,\phi} = I$ with parameters $\phi$ and $\theta$. Define $\mathbf{F}_{j_1|j_2,\phi} := \mathbf{F}_{j_1,\phi} \circ \cdots \circ \mathbf{F}_{j_2,\phi}$ by $\mathbf{F}_{j,\phi} : \mathbb{D}(V_{-j}) \mapsto \mathbb{D}(V_{-j+1})$ where $\mathbf{F}_{j,\phi} := proj_{V_{-j+1}} \circ F_{j,\phi}$, and analogously define $\mathbf{B}_{j_1|j_2,\theta}$ with $\mathbf{B}_{j,\theta} := B_{j,\theta} \circ embd_{V_{-j}}$. Then, the sequence $\{\mathbf{B}_{1|j,\theta}(\mathbf{F}_{1|J,\phi}\nu_J)\}_{j=0}^J$ forms a discrete multi-resolution bridge between $\mathbf{F}_{1|J,\phi}\nu_J$ and $\mathbf{B}_{1|J,\theta}\mathbf{F}_{1|J,\phi}\nu_J$ at times $\{t_j\}_{j=1}^J$, and*

$$
\sum_{j=0}^J \mathbb{E}_{X_{t_j} \sim \nu_j} \left\| proj_{U_{-j+1}} X_{t_j} \right\|_2^2 \Big/ \left\| \mathbf{F}_{j|J,\phi} \right\|_2^2 \leq (\mathcal{W}_2(\mathbf{B}_{1|J,\theta}\mathbf{F}_{1|J,\phi}\nu_J, \nu_J))^2, \tag{3}
$$

*where $\mathcal{W}_2$ is the Wasserstein-2 metric and $\left\| \mathbf{F}_{j|J,\phi} \right\|_2$ is the Lipschitz constant of $\mathbf{F}_{j|J,\phi}$.*

Theorem 3 states that the bottleneck component of a U-Net pushes the latent data distribution to a finite multi-resolution basis, specifically a Haar basis when average pooling is used. To see this, note that the RHS of Eq. (A.65) is itself upper-bounded by the $L^2$-reconstruction error. This is because the Wasserstein-2 distance finds the infimum over all possible couplings between the data and the 'reconstruction' measure, hence any coupling (induced by the learned model) bounds it. Note that models using a U-Net, for instance HVAEs or diffusion models, either directly or indirectly optimise for low reconstruction error in their loss function. The LHS of Eq. (A.65) represents what percentage of our data enters the orthogonal subspaces $\{U_{-j}\}_{j=0}^J$ which are (by Theorem 2) discarded by the bottleneck structure when using a U-Net architecture with average pooling. Theorem 3 thus shows that as we minimise the reconstruction error during training, we minimise the percentage of our data transported to the orthogonal sub-spaces $\{U_{-j}\}_{j=0}^J$. Consequently, the bottleneck architecture implicitly decomposes our data into a Haar wavelet decomposition, and when the skip connections are absent (like in a traditional auto-encoder) our network learns to compress the discarded subspaces $U_{-j}$. This characterises the regularisation imposed by a U-Net in the absence of skip connections.

These results suggest that U-Nets with average pooling provide a direct alternative to Fourier features [19, 33, 34, 35] which impose a Fourier basis, an alternative orthogonal basis on $L^2(\mathbb{X})$, as with skip connections the U-Net adds each subspace $U_{-j}$ sequentially. However, unlike Fourier bases, there are in fact a multitude of wavelet bases which are all encompassed by the multi-resolution framework, and in particular, Theorem 3 pertains to all of them for the bottleneck structure. This opens the door to exploring conjugacy operations beyond average pooling induced by other wavelet bases optimised for specific data types.

### 2.3 Example: HVAEs as Diffusion Discretisations

To show what practical inferences we can derive from our multi-resolution framework, we apply it to analyse state-of-the-art HVAE architectures (see Appendix B.3 for an introduction), identifying parameter redundancies and instabilities. Here and in our experiments, we focus on VDVAEs [9]. We provide similar results for Markovian HVAEs [36, 37] and NVAEs [10] (see § 4) in Appendix A.5.

We start by inspecting VDVAEs. As we show next, we can tie the computations in VDVAE cells to the (forward and backward) operators $F_{j,\phi}$ and $B_{j,\theta}$ within our framework and identify them as a type of two-step forward Euler discretisation of a diffusion process. When used with a U-Net, as is done in VDVAE [9], this creates a *multi-resolution diffusion bridge* by Theorem 4.

**Theorem 4.** *Let $t_J := T \in (0,1)$ and consider (the $p_\theta$ backward pass) $\boldsymbol{B}_{\theta,1|J} : \mathbb{D}(V_{-J}) \mapsto \mathbb{D}(V_0)$ given in multi-resolution Markov process in the standard basis:*

$$dZ_t = (\overleftarrow{\mu}_{1,t}(Z_t) + \overleftarrow{\mu}_{2,t}(Z_t))dt + \overleftarrow{\sigma}_t(Z_t)dW_t, \tag{4}$$

*where $proj_{U_{-j}} Z_{t_j} = 0$, $\|Z_t\|_2 > \|Z_s\|_2$ with $0 \le s < t \le T$ and for a measure $\nu_J \in \mathbb{D}(V_{-J})$ we have $X_T$, $Z_0 \sim \boldsymbol{F}_{\phi,J|1}\nu_J = \delta_{\{0\}}$. Then, VDVAEs approximates this process, and its residual cells are a type of two-step forward Euler discretisation of this Stochastic Differential Equation (SDE).*

To better understand Theorem 4, we visualise its residual cell structure of VDVAEs and the corresponding discretisation steps in Fig. 4, and together those of NVAEs and Markovian HVAEs in Appendix A.5, Fig. A.1. Note that this process is Markov and increasing in the $Z_i$ variables. Similar processes have been empirically observed as efficient first-order approximates to higher-order chains, for example the memory state in LSTMs [38]. Further, VDVAEs and NVAEs are even claimed to be high-order chains (see Eqs. (2,3) in [9] and Eq. (1) in [10]), despite only approximating this with a accumulative process.

To show how VDVAEs impose the growth of the $Z_t$, we prove that the bottleneck component of VDVAE's U-Net enforces $Z_0 = 0$. This is done by identifying that the measure $\nu_0$, which a VDVAE connects to the data $\nu_\infty$ via a multi-resolution bridge, is a point mass on the zero function. Consequently the backward pass must grow from this, and the network learns this in a monotonic manner as we later confirm in our experiments (see §3.2).

**Theorem 5.** *Consider the SDE in Eq. (A.76), trained through the ELBO in Eq. B.101. Let $\tilde{\nu}_J$ denote the data measure and $\nu_0 = \delta_{\{0\}}$ be the initial multi-resolution bridge measure imposed by VDVAEs. If $q_{\phi,j}$ and $p_{\theta,j}$ are the densities of $B_{\phi,1|j}\boldsymbol{F}_{J|1}\tilde{\nu}_J$ and $B_{\theta,1|j}\nu_0$ respectively, then a VDVAE optimises the boundary condition $\min_{\theta,\phi} KL(q_{\phi,0,1}\|q_{\phi,0}p_{\theta,1})$, where a double index indicates the joint distribution.*

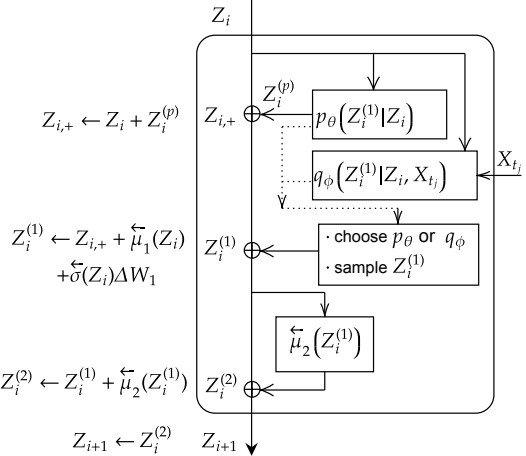

Figure 4: The VDVAE [9] cell is a type of two-step forward Euler discretisations of the continuous-time diffusion process in Eq. A.76. See Fig. A.1 for similar schemas on NVAE [10] and Markovian HVAE [36, 37].

Theorem 5 states that the VDVAE architecture forms multi-resolution bridge with the dynamics of Eq. (A.76), and connects our data distribution to the trivial measure on $V_0$: a Dirac mass at $0$ as the pooling here cascades completely to $V_0$. From this insight, we can draw conclusions on instabilities and on parameter redundancies of this HVAE cell. There are two major instabilities in this discretisation. First, the imposed $\nu_0$ is disastrously unstable as it enforces a data set, with potentially complicated topology to derive from a point-mass in $U_{-j}$ at each $t = t_j$, and we observe the resulting sampling instability in our experiments in §3.3. We note that similar arguments are applicable in settings without a latent hierarchy imposed by a U-Net, see for instance [39]. The VDVAE architecture does, however, bolster this rate through the $Z_{i,+}^{(\sigma)}$ term, which is absent in NVAEs [10], in the discretisation steps of the residual cell. We empirically observe this controlled backward error in Fig. 6 [Right]. We refer to Fig. A.1 for a detailed comparison of HVAE cells and their corresponding discretisation of the coupled SDE in Eq. (A.76).

Moreover, the current form of VDVAEs is over-parameterised and not informed by this continuous-time formulation. The continuous time analogue of VDVAEs [9] in Theorem 4 has time dependent coefficients $\overleftarrow{\mu}_{t,1}, \overleftarrow{\mu}_{t,2}, \overleftarrow{\sigma}_t$. We hypothesise that the increasing diffusion process in $Z_i$ implicitly encodes time. Hence, explicitly representing this in the model, for instance via ResNet blocks with independent parameterisations at every time step, is redundant, and a time-homogeneous model (see Appendix A.6 for a precise formulation)—practically speaking, performing weight-sharing across time time steps/layers—has the same expressivity, but requires far fewer parameters than the state-of-the-art VDVAE. It is worth noting that such a time-homogeneous model would make the parameterisation of HVAEs more similar to the recently popular (score-based) diffusion models [40, 41] which perform weight-sharing across all time steps.

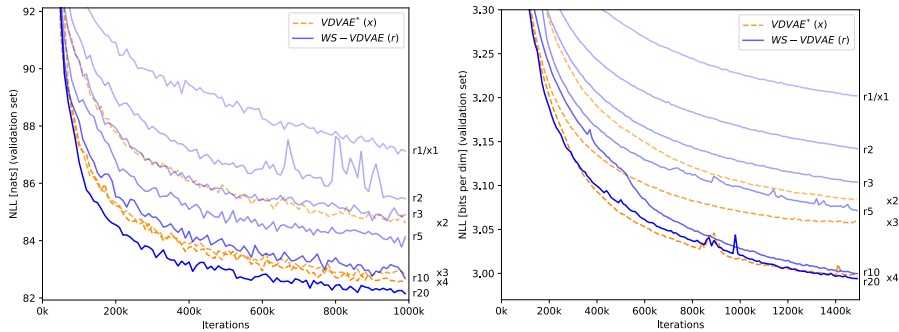

Figure 5: A small-scale study on parameter efficiency of HVAEs. We compare models with with 1,2,3 and 4 parameterised blocks per resolution ($\{x1, x2, x3, x4\}$) against models with a single parameterised block per resolution weight-shared $\{2, 3, 5, 10, 20\}$ times ($\{r2, r3, r5, r10, r20\}$). We report NLL ($\downarrow$) measured on the validation set of MNIST [left] and CIFAR10 [right]. NLL performance increases with more weight-sharing repetitions and surpasses models without weight-sharing but with more parameters.

## 3 Experiments

In the following we probe the theoretical understanding of HVAEs gained through our framework, demonstrating its utility in four experimental analyses: (*a*) Improving parameter efficiency in HVAEs, (*b*) Time representation in HVAEs and how they make use of it, (*c*) Sampling instabilities in HVAEs, and (*d*) Ablation studies.

We train HVAEs using VD-VAE [9] as the basis model on five datasets: MNIST [42], CI-FAR10 [43], two downsampled versions of ImageNet [44, 45], and CelebA [46], splitting each into a training, validation and test set (see Appendix D for details). In general, reported numeric values refer to Negative Log-Likelihood (NLL) in nats (MNIST) or bits per dim (all other datasets) on the test set at model convergence, if not stated otherwise. We note that performance on the validation and test set have similar trends in general. An optional *gradient checkpointing* implementation to trade in GPU memory for compute is discussed in Appendices F. Appendices F and G define the HVAE models we train, i.e. $p_\theta(\mathbf{z}_L), p_\theta(\mathbf{z}_l | \mathbf{z}_{>l}), q_\phi(\mathbf{z}_L | \mathbf{x}), q_\phi(\mathbf{z}_l | \mathbf{z}_{>l}, \mathbf{x})$ and $p_\theta(\mathbf{x} | \vec{\mathbf{z}})$, and present additional experimental details and results. We provide our PyTorch code base at https://github.com/FabianFalck/unet-vdvae (see Appendix C for details).

Table 1: A large-scale study of parameter efficiency in HVAEs. We compare our runs of VDVAE with original hyperparameters [9] (VDVAE*) against our weight-shared VDVAE (WS-VDVAE). While WS-VDVAEs have improved parameter efficiency by a factor of 2, they reach similar NLL as VDVAE* with the simple modification inspired by our framework (weight sharing). We note that a parameter count cannot be provided for VDM [19] as the code is not public and the manuscript does not specify it.

| Dataset | Method | Type | #Params | NLL ↓ |
|---|---|---|---|---|
| **MNIST** 28×28 | WS-VDVAE (ours) | VAE | **232k** | ≤ 79.98 |
| | VDVAE* (ours) | VAE | 339k | ≤ 80.14 |
| | NVAE [10] | VAE | 33m | ≤ 78.01 |
| **CIFAR10** 32×32 | WS-VDVAE (ours) | VAE | **25m** | ≤ 2.88 |
| | WS-VDVAE (ours) | VAE | 39m | ≤ 2.83 |
| | VDVAE* (ours) | VAE | 39m | ≤ 2.87 |
| | NVAE [10] | VAE | 131m | ≤ 2.91 |
| | VDVAE [9] | VAE | 39m | ≤ 2.87 |
| | VDM [19] | Diff | – | ≤ 2.65 |
| **ImageNet** 32×32 | WS-VDVAE (ours) | VAE | **55m** | ≤ 3.68 |
| | WS-VDVAE (ours) | VAE | 85m | ≤ 3.65 |
| | VDVAE* (ours) | VAE | 119m | ≤ 3.67 |
| | NVAE [10] | VAE | 268m | ≤ 3.92 |
| | VDVAE [9] | VAE | 119m | ≤ 3.80 |
| | VDM [19] | Diff | – | ≤ 3.72 |
| **CelebA** 64×64 | WS-VDVAE (ours) | VAE | **75m** | ≤ 2.02 |
| | VDVAE* (ours) | VAE | 125m | ≤ 2.02 |
| | NVAE [10] | VAE | 153m | ≤ 2.03 |

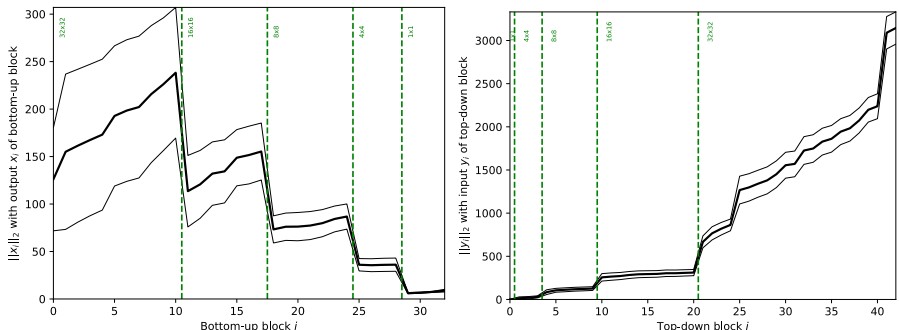

Figure 6: HVAEs secretly represent a notion of time: We measure the $L_2$-norm of the residual state for the [Left] forward/bottom-up pass and the [Right] backward/top-down pass over 10 batches with 100 data points each. In both plots, the thick, central line refers to the average and the thin, outer lines refer to $\pm 2$ standard deviations.

## 3.1 "More from less": Improving parameter efficiency in HVAEs

In §2.3, we hypothesised that a time-homogeneous model has the same expressivity as a model with time-dependent coefficients, yet uses much less parameters. We start demonstrating this effect by weight-sharing ResNet blocks across time on a small scale. In Fig. 5, we train HVAEs on MNIST and CIFAR10 with $\{1, 2, 3, 4\}$ ResNet blocks (referred to as $\{$x1, x2, x3, x4$\}$) in each resolution with spatial dimensions $\{32^2, 16^2, 8^2, 4^2, 1^2\}$ (VDVAE*), and compare their performance when weight-sharing a single parameterised block per resolution $\{2, 3, 5, 10, 20\}$ times (referred to as $\{$r2,r3,r5,r10,r20$\}$; WS-VDVAE), excluding projection and embedding blocks. As hypothesised by our framework, yet very surprising in HVAEs, NLL after 1m iterations measured on the validation set gradually increases the more often blocks are repeated even though all weight-sharing models have an identical parameter count to the $x1$ model (MNIST: 107k, CIFAR10: 8.7m). Furthermore, the weight-sharing models often outperform or reach equal NLLs compared to x2, x3, x4, all of which have more parameters (MNIST: 140k; 173k; 206k. CIFAR10: 13.0m; 17.3m; 21.6m), yet fewer activations, latent variables, and number of timesteps at which the coupled SDE in Eq. (A.76) is discretised.

We now scale these findings up to large-scale hyperparameter configurations. We train VDVAE closely following the state-of-the-art hyperparameter configurations in [9], specifically with the same number of parameterised blocks and without weight-sharing (VDVAE*), and compare them against models with weight-sharing (WS-VDVAE) and fewer parameters, i.e. fewer parameterised blocks, in Table 1. On all four datasets, the weight-shared models achieve similar NLLs with fewer parameters compared to their counterparts without weight-sharing: We use $32\%$, $36\%$, $54\%$, and $40\%$ less parameters on the four datasets reported in Table 1, respectively. For the larger runs, weight-sharing has diminishing returns on NLL as these already have many discretisation steps. To the best of our knowledge, our models achieve a new state-of-the-art performance in terms of NLL compared to any HVAE on CIFAR10, ImageNet32 and CelebA. Furthermore, our WS-VDVAE models have stochastic depths of 57, 105, 235, 125, respectively, the highest ever trained. In spite of these results, it is worth noting that current HVAEs, and VDVAE in particular remains notoriously unstable to train, partly due to the instabilities identified in Theorem 5, and finding the right hyperparameters helps, but cannot solve this.

## 3.2 HVAEs secretly represent time and make use of it

In §3.1, we showed how we can exploit insight on HVAEs through our framework to make HVAEs more parameter efficient. We now want to explain and understand this behavior further. In Fig. 6, we measure $\|Z_i\|_2$, the $L_2$-norm of the residual state at every backward/top-down block with index i, over several batches for models trained on MNIST (see Appendix G.2 for the corresponding figure of the forward/bottom-up pass, and similar results on CIFAR10 and ImageNet32). On average, we experience an increase in the state norm across time in every resolution, interleaved by discontinuous 'jumps' at the resolution transitions (projection or embedding) where the dimension of the residual state changes. This supports our claim in §2 that HVAEs discretise multi-resolution diffusion

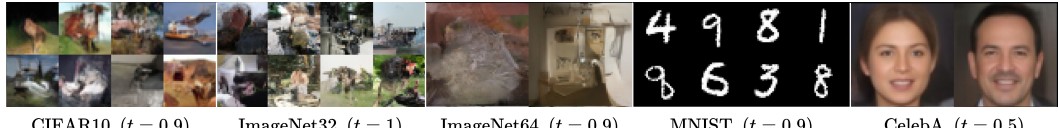

| CIFAR10 ($t = 0.9$) | ImageNet32 ($t = 1$) | ImageNet64 ($t = 0.9$) | MNIST ($t = 0.9$) | CelebA ($t = 0.5$) |

Figure 7: Unconditional samples (not cherry-picked) of VDVAE*. While samples on MNIST and CelebA demonstrate high fidelity and diversity, samples on CIFAR10, ImageNet32 and ImageNet64 are diverse, but are unrecognisable, demonstrating the instabilities identified by Theorem 1. Temperatures $t$ are tuned for maximum fidelity.

processes which are increasing in the $Z_i$ variables, and hence learn to represent a notion of time in their residual state.

It is now straightforward to ask how HVAEs benefit from this time representation during training: As we show in Table 2, when normalising the state by its norm at every forward and backward block during training, i.e. forcing a "flat line" in Fig. 6 [Left], learning deteriorates after a short while, resulting in poor NLL results compared to the runs with a regular, non-normalised residual state. This evidence confirms our earlier stated hypothesis: The time representation in ResNet-based HVAEs encodes information which recent HVAEs heavily rely on during learning.

### 3.3 Sampling instabilities in HVAEs

High fidelity unconditional samples of faces, e.g. from models trained on CelebA, cover the front pages of state-of-the-art HVAE papers [9, 10]. Here, we question whether face datasets are an appropriate benchmark for HVAEs. In Theorem 5, we identified the aforementioned state-of-the-art HVAEs as flow from a point mass, hypothesising instabilities during sampling. And indeed, when sampling from our trained VDVAE* with state-of-the-art configurations, we observe high fidelity and diversity samples on MNIST and CelebA, but unrecognisable, yet diverse samples on CIFAR10, ImageNet32 and ImageNet64, in spite of state-of-the-art test set NLLs (see Fig. 7 and Appendix G.3). We argue that MNIST and CelebA, i.e. numbers and faces, have a more uni-modal nature, and are in this sense easier to learn for a discretised multi-resolution process flowing to a point mass, which is uni-modal, than the other "in-the-wild", multi-modal datasets. Trying to approximate the latter with the, in this case unsuitable, HVAE model leads to the sampling instabilities observed.

Table 2: NLL of HVAEs with and without normalisation of the residual state $Z_i$.

| Residual state | NLL |
|---|---|
| **MNIST** | |
| Normalised (✗) | $\leq 464.68$ |
| Non-normalised | $\leq 81.69$ |
| **CIFAR10** | |
| Normalised (✗) | $\leq 6.80$ |
| Non-normalised | $\leq 2.93$ |
| **ImageNet** | |
| Normalised | $\leq 6.76$ |
| Non-normalised | $\leq 3.68$ |

### 3.4 Ablation studies

We conducted several ablation studies which support our experimental results and further probe our multi-resolution framework for HVAEs. In this section we note key findings—a detailed account of all ablations can be found in Appendix G.4. In particular, we find that the number of latent variables, which correlates with stochastic depth, does not explain the performance observed in §3.1, supporting our claims. We further show that Fourier features do not provide a performance gain in HVAEs, in contrast to state-of-the-art diffusion models, where they significantly improve performance [19]. This is consistent with our framework's finding that a U-Net architecture with pooling is already forced to learn a Haar wavelet basis representation of the data, hence introducing another basis does not add value. We also demonstrate that residual cells are crucial for the performance of HVAEs as they are able to approximate the dynamics of a diffusion process and impose an SDE structure into the model, empirically compare a multi-resolution bridge to a single-resolution model, and investigate synchronous vs. asynchronous processing in time between the forward and backward pass.

# 4 Related work

**U-Nets.** A U-Net [1] is an autoencoding architecture with multiple resolutions where skip connections enable information to pass between matched layers on opposite sides of the autoencoder's bottleneck. These connections also smooth out the network's loss landscape [47]. In the literature, U-Nets tend to be convolutional, and a wide range of different approaches have been used for up-sampling and down-sampling between resolutions, with many using average pooling for the down-sampling operation [13, 14, 16, 17, 19]. In this work, we focus on U-Nets as operators on measures interleaved by average pooling as the down-sampling operation (and a corresponding inclusion operation for up-sampling), and we formally characterise U-Nets in Section 2.1 and Appendix B.2. Prior to our work, the dimensionality-reducing bottleneck structure of U-Nets was widely acknowledged as being useful, however it was unclear what regularising properties a U-Net imposes. We provided these in §2.

**HVAEs.** The evolution of HVAEs can be seen as a quest for a parameterisation with more expressiveness than single-latent-layer VAEs [48], while achieving stable training dynamics that avoid common issues such as posterior collapse [36, 49] or exploding gradients. Early HVAEs such as LVAE condition each latent variable directly on only the previous one by taking samples forward [36, 37]. Such VAEs suffer from stability issues even for very small stochastic depths. *Nouveau VAEs (NVAE)* [10] and *Very Deep VAEs (VDVAE)* [9] combine the improvements of several earlier HVAE models (see Appendix B for details), while scaling up to larger stochastic depths. Both use ResNet-based backbones, sharing parameters between the generative and recognition parts of the model. VDVAE is the considerably simpler approach, in particular avoiding common tricks such as a warm-up deterministic autoencoder training phase or data-specific initialisation. VDVAE achieves a stochastic depth of up to 78, improving performance with more ResNet blocks. Worth noting is that while LVAE and NVAE use convolutions with appropriately chosen stride to jump between resolutions, VDVAE use average pooling. In all HVAEs to date, a theoretical underpinning which explains architectural choices, for instance the choice of residual cell, is missing, and we provided this in Section §2.3.

# 5 Conclusion

In this work, we introduced a multi-resolution framework for U-Nets. We provided theoretical results which uncover the regularisation property of the U-Nets bottleneck architecture with average pooling as implicitly learning a Haar wavelet representation of the data. We applied our framework to HVAEs, identifying them as multi-resolution diffusion processes flowing to a point mass. We characterised their backward cell as a type of two-step forward Euler discretisations, providing an alternative to score-matching to approoximate a continuous-time diffusion process [16, 18], and observed parameter redundancies and instabilities. We verified the latter theoretical insights in both small- and large-scale experiments, and in doing so trained the deepest ever HVAEs. We explained these results by showing that HVAEs learn a representation of time and performed extensive ablation studies.

An important limitation is that the proven regularisation property of U-Nets is limited to using average pooling as the down-sampling operation. Another limitation is that we only applied our framework to HVAEs, though it is possible to apply it to other model classes. It could also be argued that the lack of exhaustive hyperparameter optimisation performed is a limitation of the work as it may be possible to obtain improved results. We demonstrate, however, that simply adding weight-sharing to the hyperparameter settings given in the original VDVAE paper [9] leads to state-of-the-art performance with improved parameter efficiency, and hence view it as a strength of our results.

## Acknowledgments and Disclosure of Funding

Fabian Falck acknowledges the receipt of studentship awards from the Health Data Research UK-The Alan Turing Institute Wellcome PhD Programme in Health Data Science (Grant Ref: 218529/Z/19/Z), and the Enrichment Scheme of The Alan Turing Institute under the EPSRC Grant EP/N510129/1. Chris Williams acknowledges support from the Defence Science and Technology (DST) Group and from a ESPRC DTP Studentship. Dominic Danks is supported by a Doctoral Studentship from The Alan Turing Institute under the EPSRC Grant EP/N510129/1. Christopher Yau is funded by a UKRI Turing AI Fellowship (Ref: EP/V023233/1). Chris Holmes acknowledges support from the Medical Research Council Programme Leaders award MC_UP_A390_1107, The Alan Turing Institute, Health Data Research, U.K., and the U.K. Engineering and Physical Sciences Research Council through the Bayes4Health programme grant. Arnaud Doucet acknowledges support of the UK Defence Science and Technology Laboratory (Dstl) and EPSRC grant EP/R013616/1. This is part of the collaboration between US DOD, UK MOD and UK EPSRC under the Multidisciplinary University Research Initiative. Arnaud Doucet also acknowledges support from the EPSRC grant EP/R034710/1. Matthew Willetts is grateful for the support of UCL Computer Science and The Alan Turing Institute.

The authors report no competing interests.

The three compute clusters used in this work were provided by the Alan Turing Institute, the Oxford Biomedical Research Computing (BMRC) facility, and the Baskerville Tier 2 HPC service (https://www.baskerville.ac.uk/) which we detail in the following. First, this research was supported in part through computational resources provided by The Alan Turing Institute under EPSRC grant EP/N510129/1 and with the help of a generous gift from Microsoft Corporation. Second, we used the Oxford BMRC facility, a joint development between the Wellcome Centre for Human Genetics and the Big Data Institute supported by Health Data Research UK and the NIHR Oxford Biomedical Research Centre. The views expressed are those of the author(s) and not necessarily those of the NHS, the NIHR or the Department of Health. Third, Baskerville was funded by the EPSRC and UKRI through the World Class Labs scheme (EP/T022221/1) and the Digital Research Infrastructure programme (EP/W032244/1) and is operated by Advanced Research Computing at the University of Birmingham.

We thank Tomas Lazauskas, Jim Madge and Oscar Giles from the Alan Turing Institute's Research Engineering team for their help and support. We thank Adam Huffman, Jonathan Diprose, Geoffrey Ferrari and Colin Freeman from the Biomedical Research Computing team at the University of Oxford for their help and support. We thank Haoting Zhang (University of Cambridge) for valuable comments on the implementation; Huiyu Wang (Johns Hopkins University) for a useful discussion on gradient checkpointing; and Ruining Li and Hanwen Zhu (University of Oxford) for kindly proofreading the manuscript.

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
