# OpenReview forum: "A Multi-Resolution Framework for U-Nets with Applications to Hierarchical VAEs"
_NeurIPS.cc/2022/Conference — NeurIPS 2022 Accept_

### Official Review · Reviewer_yjaZ · 2022-06-28

**Rating:** 7
**Confidence:** 2
**Soundness:** 3 good
**Presentation:** 3 good
**Contribution:** 3 good

**Summary:**

This paper proposes a multi-resolution framework of U-net. Theoretical results show the regularisation property of a U-Net with average pooling as implicitly learning a Haar wavelet representation of the data. This framework is also used to identify parameter redundancies and instabilities of HVAEs and verify them via experiments.

**Questions:**

No.

**Ethics Review Area:**

["I don’t know"]

**Limitations:**

(1) Only test on a single VAE model HVAEs.

**Strengths And Weaknesses:**

Pros:

(1) This is a paper with theoritical insights that help to understand the regularization properties of U-net.

(2) With the help of the proposed theories, the parameter efficiencies of HVAEs are improved as shown by extensive experiments on 4 benchmarks.

Cons:

(1) As admitted in the paper, the author only applies the framework to a single VAE model HVAEs so the theory is not guaranteed to generalize.

(2) Though the theory is insightful and novel, the practical technique used to improve parameter efficiencies is just parameter sharing, which is well studied.

---

> ### Author Response · Authors · 2022-08-02
> **Response to Reviewer yjaZ (1 of 2)**
>
> Thank you for your thoughtful and encouraging review.
> We thank you for appreciating the theoretical contribution of our work, uncovering the regularisation properties of U-Nets, as well as the application of our framework to HVAEs in extensive experiments.
> We are also glad you like the insight and novelty of our theory, and thank you for your positive score!
> Below, we would like to address the two points mentioned as weaknesses of our work.
>
> - - -
>
> > “As admitted in the paper, the author only applies the framework to a single VAE model HVAEs so the theory is not guaranteed to generalize.”
> > “Only test on a single VAE model HVAEs.”
>
> It is important to us to clarify this point:
>
> We agree that Theorems 4 and 5 are pertaining to VDVAE [4], a particular instance of HVAEs, however, this is a rather exemplary choice for the exposition of the main text and for consistency with our experiments. Our framework is not limited to VDVAE, it is likewise applicable to other HVAEs (namely NVAE [3], Markovian HVAEs [1,2]) which satisfy the assumptions of the theorems, and we have provided evidence and further analysis for this in the supplementary of our submission.
>
> With regards to Theorem 4, in Appendix A.5, Fig. A.1, we have analysed the HVAE cells of Markovian HVAEs [1,2] and NVAE [3], two other, important HVAE instances, and precisely demonstrate how their internal mechanism corresponds to a two-step forward Euler discretisation of the continuous-time diffusion process represented via the coupled SDEs in the theorem. We referred to these additional results in l. 184 and 196 in the main text, but based on your feedback, we now point there to the exact Subsection and Figure to make clear that we have demonstrated our framework broadly on HVAEs.
>
> It is also important to note that Theorems 1 to 3 and corresponding Lemmas and Propositions in Appendix A pertain to U-Nets in general, and are hence applicable to many other generative model classes which use a U-Net, for instance score-based models or U-Nets used in image segmentation.
>
> While we are not sure, the confusion may be referring to this sentence from our submission when stating the above weakness: “Another limitation is that we only applied our framework to HVAEs, though it is possible to apply it to other model classes.” We apologise if this sentence was misleading, and would like to clarify: What we mean is that Theorems 4 and 5 are applications of our framework limited to HVAEs (Markovian HVAEs, NVAE, VDVAE), identifying them as forward Euler discretisations of multi-resolution diffusion processes. With “other model classes'' we meant for instance score-based diffusion models, an alternative to HVAEs for approximating a diffusion process, which could similarly be designed and analysed within our multi-resolution framework. While this is beyond the scope of our work, it highlights the potential exciting avenues that our novel framework enables. We again thank you for pointing this potential misunderstanding out, and make the following change of this sentence: “[...]Another limitation is that we only applied our framework to HVAEs, though it is possible to apply it to other model classes, *such as score-based diffusion models*.”

---

> > ### Author Response · Authors · 2022-08-02
> > **Response to Reviewer yjaZ (2 of 2)**
> >
> > > “Though the theory is insightful and novel, the practical technique used to improve parameter efficiencies is just parameter sharing, which is well studied.”
> >
> > We agree that weight-sharing itself is a well-studied phenomenon. In the context of HVAEs, however, it has not been studied at all, either from a theoretical or empirical perspective. Further, it is surprising that it yields any empirical benefit, and our theoretical advances do explain how weight-sharing HVAEs operate. To the best of our knowledge, this is the first time that HVAEs have been shown to benefit—across small- and large-scale experiments—from weight-sharing. It also drastically improves parameter efficiency of the current state-of-the-art performance.
> >
> > It is important to stress that this identification of a parameter redundancy was only enabled by our framework.
> > First, the identification of HVAEs as forward Euler discretisations of multi-resolution, forward and backward diffusion processes in Theorem 4 was key to have the intuition of thinking about their numerical backward error $\epsilon_i$. We then hypothesised that one way to control $\epsilon_i$ is through an increasing norm of $Y_i$, the backward residual state, which we now measured. Again, it is very surprising to empirically observe this phenomenon of an increasing $\| \| Y_i \| \|_2$ in Figure 3 [Left].
> >
> > Only based on this theoretical insight and empirical validation, we hypothesised that $\| \| Y_i \| \|_2$ may capture some notion of time that our HVAE exploits during training. And indeed, when we normalised $Y_i$ by its $L_2$-norm in Section 3.2, we observed deteriorated results. This motivated us to consider a time-homogeneous diffusion (see Appendix A.8 for a precise formulation) where the parameterisations (neural nets) of the diffusion coefficients are shared across time, because time is already captured within their input, hence had the idea of weight-sharing across time.
> >
> > We would like to emphasise that improving the parameter efficiency of HVAEs is by far not the only practical impact our theory has: In our paper, beyond the insights described above, our theory enabled to practically understand and measure how HVAEs with a many latent layers (=stochastic depth) are stabilised, a phenomenon which was previously only empirically observed [3,4]. For HVAEs, this will have immediate practical consequences, such as—based on the insight of our framework (Theorem 4)---to invent HVAEs corresponding to better discretisations of the coupled SDEs in Eq. (2), which are designed based on a clear theoretical motivation in contrast to all previous HVAEs, and may in addition be more performant and stable. Our framework could also have very practical impacts on score-based diffusion models and many other model classes which likewise use U-Net architectures as their go-to parameterisations of choice. Our framework is a first step towards understanding U-Nets, a key building block in state-of-the-art generative modelling. And we hope you agree with us that our work is for these reasons worth presenting to the wider NeurIPS community.

---

> > > ### Comment · Reviewer_yjaZ · 2022-08-07
> > > **My concerns are addressed**
> > >
> > > Thanks for your clarification. I agree with the authors that this work is also applicable to our VAE models. I also agree that this is the first time that weight sharing is studied under a VAE framework. I'd like to improve my score to 7.

---

> > > > ### Author Response · Authors · 2022-08-08
> > > > **Author response to “My concerns are addressed”**
> > > >
> > > > Thank you very much for your response! We appreciate discussing and clarifying these two points on the contribution of our work, and thank you very much for increasing your score to an “accept” to reflect this!

---

### Official Review · Reviewer_sqR1 · 2022-07-11

**Rating:** 5
**Confidence:** 2
**Soundness:** 2 fair
**Presentation:** 1 poor
**Contribution:** 2 fair

**Summary:**

This work tries to make an abstraction of the UNet (through the multi-resolution framework). Based on this abstraction, this work analyzes properties of UNet, revealing a regularisation property of UNet, that is implicitly learning a Haar wavelet basis representation. This work further identifies HVAE (based on UNet) as an Euler discretisation of multi-resolution diffusion sum processes. This paper also proposes a weight sharing of HVAE, which reduces the number of parameters.

**Questions:**



**Limitations:**



**Strengths And Weaknesses:**

Overall, the analysis framework of multi-resolution is original from my perspective, but I preserve some questions on this framework. The paper builds upon heavy notations with less explanation. Therefore, I think the writting quality and clarity can be improved. Besides, it seems that the UNet used in practice has a gap between the UNet discussed in this work, and this gap is not discussed clearly.

Below are my concerns and questions in detail:

1. The multi-resolution framework use finite dimensional projection. So I expect a projection is defined on a discrete space, such as $Z^2$. However, the partition is w.r.t. functions with domain $R^2$ in Definition 1.

2. In definition 1, why a negative $j$ corresponds to zooming in? Suppose $f(\cdot) \in V_j$, where $j$ is negative, and the input space $[0, 2^j]^2$. Such an input space is scaled to $[0, 1]^2$ for $f(2^j \cdot)$ to keep the same graph. Therefore, a negative $j$ has a smaller graph (since $2^j < 1$) compared to $j=0$. From my perspective, a smaller graph corresponds to zooming out.

3. In definition 2, the author mentions the multi-resolution partition of $L^2(X)$, where $X$ is a compact set. But the partition on a compact set is not defined. It seems that directly applying definition 1 to compact set will have some problems, since A4 implies $X$ should be unbounded to allow a translation transformation. However, a compact is bounded. Therefore, I think it is necessary to define the multi-resolution partition for a compact set $X$ in some place.

4. This paper does not give an introduction to the UNet. This makes it difficult to link the UNet used in practice to the one discussed in this paper. The author could introduce the UNet used in practice somewhere and discuss how the practical UNet is abstracted in this paper.

5. The practical UNet has different number of channels in different layers. Does this work also support the different number of channels?

6. An important feature of UNet is the skip connection across different layers. It seems that the framework in this paper ignores this feature.

7. This paper relies on heavy notations. These notations should be explained more. For example, the author could explain what the practical counterpart is when introducing a notation.

8. When the author refers to the Appendix, the author could refer to a more detailed position.

9. This paper relies on UNet, HVAE and multi-resolution partition. It would be better to add a background section to introduce these important prior works. This would make this paper more readable.

---

> ### Author Response · Authors · 2022-08-02
> **Response to Reviewer sqR1 (1 of 5)**
>
> We thank you for your detailed and thoughtful review and are pleased to see that you recognise our framework as original. We respectfully disagree on some of the potential concerns you have raised. We would like to address these together with your questions below and have made changes in the revision to accommodate them. We would appreciate it if you would consider our response towards your final evaluation of the submission.
>
>
> - - -
>
> > “1. The multi-resolution framework use finite dimensional projection. So I expect a projection is defined on a discrete space, such as $\mathbb{Z}^2$. However, the partition is w.r.t. functions with domain $\mathbb{R}^2$ in Definition 1.”
>
> We respectfully disagree that defining a finite-dimensional projection on $\mathbb{Z}^2$ would be useful for our framework, nor is it standard in similar contexts.
> Instead, we argue that defining it on $\mathbb{R}^2$ as we have done is beneficial and correct.
> We elaborate on this below.
>
> Our framework defines projections to a finite function space, where the domain of each function is $\mathbb{R}^2$, not $\mathbb{Z}^2$.
> This definition of a finite projection is very natural in other contexts, for instance in Fourier analysis:
> When considering a Fourier basis for ($L^2$) functions on $\mathbb{R}^2$, a finite projection of a function means taking the basis elements corresponding to its first $n$ Fourier coefficients.
> Importantly, each Fourier basis function is defined on the whole of $\mathbb{R}^2$, its domain is *not* changing.
> In our framework, we do precisely this, but with the first $n$ many wavelet basis elements, all of which are defined on $\mathbb{R}^2$.
> Another way of saying this is that—taken the context of Fourier analysis again—our finite-dimensional projection is done in the frequency domain, *not* in the time domain.
>
> Let us be more precise about this point: Our usage of the term finite-dimensional projection pertains to the *dimension* of a function space over $\mathbb{R}^2$.
> The space $\mathbb{R}^2$ itself is not shrinking, but remains the same.
> In our framework we consider approximations in the space of square integrable functions defined on $\mathbb{R}^n$.
> This space, $L^2(\mathbb{R}^n)$, is infinite dimensional.
> A multi-resolution partition gives a countable basis of $L^2(\mathbb{R}^n)$, which makes $L^2(\mathbb{R}^n)$ (like all Hilbert spaces) to be isomorphic to the space of countable square-summable real sequences.
> Hence, we may think of each function in $L^2(\mathbb{R}^n)$ as a countable sequence of real numbers, and this space is discrete in the sense asserted (in that it is countable).
> When we now only consider the first $n$ many elements of this sequence, this forms a finite dimensional truncation of $L^2(\mathbb{R}^n)$ that has dimension $n < \infty$.
> Further, as the basis we use to perform this truncation is orthonormal, this truncation is also an $L^2$-projection.
> Throughout the paper, we consider finite projections (or truncations) of exactly this form.

---

> > ### Author Response · Authors · 2022-08-02
> > **Response to Reviewer sqR1 (2 of 5)**
> >
> > > “2. In definition 1, why a negative $j$ corresponds to zooming in? Suppose $f(\cdot) \in V_j$, where $j$ is negative, and the input space $\[ 0, 2^j \]^2$. Such an input space is scaled to $\[ 0, 1 \]^2$ for $f(2^j \cdot)$ to keep the same graph. Therefore, a negative $j$ has a smaller graph (since $2^j < 1$) compared to $j=0$. From my perspective, a smaller graph corresponds to zooming out.
> >
> >
> > Thank you for your question. We appreciate that this is a subtle point and explain further below why negative $j$ corresponds to ‘zooming in’.
> > We first note that we base Definition 1 on a standard and highly cited introduction to multi-resolution analysis and wavelets in Daubechies’ [5] (1992, Section 5.1), as we state in line 81, where negative $j$ corresponds to finer detailed approximations.
> > At the same time, we thank you for this point, and acknowledge that the short sentence where we have mentioned this term could have been clearer.
> > We have made small changes to ll. 87 to accommodate this, and in a potential camera-ready version, we will use the additional page towards improving our introduction of negative $j$ as ‘zooming in’ using the explanation below.
> >
> > We respectfully disagree that in our framework the input space, or domain, of a function in $V_j$ differs from a function in $V_0$ when using the scaling property, instead, it is the *support* of the function that may change.
> > Further, the change in support pertains to our basis elements, not the function we seek to approximate, an image here, whose support is fixed.
> > In the case of our example, the image is supported on $[0,1]^2$ and our approximant lives in  $V_j$ for $j \leq 0$, each pixel is of size $2^j \cdot 2^j$ and there are $2^{-j} \cdot 2^{-j}$ many of them.
> > Put simply, the size of the ‘pixel’ changes, not the domain or support of the image itself.
> > For an illustration of this, see the picture of a cat and its finite-dimensional projections in Appendix B.1, Fig. B.2.
> >
> > For reference, the sentence where we use the term ‘zooming in’ is:
> > “In Definition 1, the index $j$ references how many steps we took in our thought experiment, so negative $j$ corresponds to `zooming in' on the images [ll. 87].”
> > We mean ‘zooming in’ in the context of our thought experiment: Here, we try to distinguish an (infinitely-detailed) image and its approximation, a finite-dimensional projection of it, both drawn on a wall.
> > Standing in front of the wall, asserting we are able to tell the two images apart, we interpreted (positive) $j$ as the number of steps taken away from the wall until the two images are indistinguishable by eye.
> > Here, $j$ is a measure of how good of an approximation the finite-dimensional projection is: The larger $j$, the worse the approximation is.
> >
> > We may also conduct the thought experiment by moving in the other direction:
> > Standing a few metres apart from the wall, asserting the two images are indistinguishable by eye, we walk towards the wall until we can distinguish the two images.
> > We will stop at the same value of $j$, with the same interpretation for it.
> > Now, we may walk towards the wall, and even when we stand right in front of it at the origin of the number line, the two images may still be indistinguishable, because the finite-dimensional projection is a very good approximation; it contains a lot of detail.
> > Then, to find $j$, which is negative relative to our origin, we need to ‘zoom in’, say with a lens, until we can see a difference.
> >
> > In summary, the illustration of ‘zooming in’ was intended to provide a physical interpretation of what a negative index $j$ means in our thought experiment.
> > We hope this resolves your concern!

---

> > > ### Author Response · Authors · 2022-08-02
> > > **Response to Reviewer sqR1 (3 of 5)**
> > >
> > > > “3. In definition 2, the author mentions the multi-resolution partition of $L^2(\mathbb{X})$, where $\mathbb{X}$ is a compact set. But the partition on a compact set is not defined. It seems that directly applying definition 1 to compact set will have some problems, since A4 implies $\mathbb{X}$ should be unbounded to allow a translation transformation. However, a compact is bounded. Therefore, I think it is necessary to define the multi-resolution partition for a compact set $\mathbb{X}$ in some place.”
> > >
> > > We agree that the link between Definition 1 and a multi-resolution partition on $L^2(\mathbb{X})$ could be made more explicit, and have included further explanation to the paper in Definition 1 to reflect this.
> > > Importantly, we state Definition 1 over $\mathbb{R}^n$, not over $\mathbb{X}$, to comply with what is standard in highly-cited literature on wavelets [5], and to exactly avoid the problems you mentioned.
> > > The link we use is the standard linear algebra construction for forming a basis on a subspace when a basis on the whole space is defined.
> > > That is, if we have a basis on the whole space, a basis on a closed linear subspace can be made by multiplying each basis element with the indicator function for the subspace.
> > > We outline this in more detail below.
> > > Thank you for this great feedback!
> > >
> > > Definition 1 gives a multi-resolution partition of $L^2(\mathbb{R}^n)$.
> > > $L^2(\mathbb{R}^n)$ has a closed linear subspace, the space of square integrable functions supported on $\mathbb{X}$.
> > > If we restrict ourselves to this subspace, it is also a Hilbert space, namely $L^2(\mathbb{X})$, and inherits the multi-resolution partition properties from $L^2(\mathbb{R}^n)$ by only considering functions in $L^2(\mathbb{R}^n)$ that have support on $\mathbb{X}$.
> > >
> > > In summary, what we propose is to first construct the multi-resolution partition over $\mathbb{R}^n$ (which is standard in the wavelet literature), then restrict it straightforwardly to $\mathbb{X}$.
> > > We agree with you that this is beneficial to first restricting to $\mathbb{X}$, then constructing the multi-resolution partition over $\mathbb{X}$, as this would result in a much more complicated, non-standard definition, which in turn may cause problems—we fully agree!
> > > We thank you for finding this subtle point and helping us improve the readability of the manuscript.
> > >
> > >
> > > - - -
> > >
> > > > “4. This paper does not give an introduction to the UNet. This makes it difficult to link the UNet used in practice to the one discussed in this paper. The author could introduce the UNet used in practice somewhere and discuss how the practical UNet is abstracted in this paper.”
> > > > “Besides, it seems that the UNet used in practice has a gap between the UNet discussed in this work, and this gap is not discussed clearly.”
> > >
> > > Thank you for this valuable feedback!
> > > We fully agree that an introduction to the U-Net—extending the short introduction in the Related work (Section 4)—particularly on the notation of our framework, the assumptions that we make, and how this notation links back with U-Nets in practice is very useful!
> > > Hence, we have prepared a new Appendix section (B.2) which shall provide this.
> > >
> > > There, Figure B.3 provides an illustration of a U-Net, using the notation of our framework.
> > > This is accompanied by extensive explanation of
> > > * the forward (encoder) and backward (decoder) neural networks $f_{\theta,j}$ and $b_{\theta,j}$,
> > > * the dimension reduction (downscaling) operation $P_{-j+1}$ between latent spaces $V_{-j}$ and $V_{-j+1}$,
> > > * the dimension embedding (upscaling) operation $E_{-j}$ between $V_{-j+1}$ and $V_{-j}$,
> > > * the potentially present second backward process with corresponding neural networks $b_{\phi,j}$,
> > > * the bottleneck
> > > * the function and importance of skip connections (see new Remark 2),
> > > * and the working definition and assumptions of an ‘idealised U-Net’ (see new Definition 4) which we use throughout our work.
> > >
> > > We would also like to state that the applicability of our framework, for instance in the form of Theorems 4 and 5, and the hypotheses generated from it, for instance on sampling instabilities and parameter redundancies, but also on numerical stability more generally and ‘representing time’,  are practically demonstrated and empirically validated on HVAEs which use a U-Net architecture.
> > > In this sense, our framework for U-Nets is directly relevant to important model classes used in practice (HVAEs), and provides an important path towards similar analyses of other model classes which use U-Nets in practice, for instance score-based diffusion models.
> > >
> > > We hope this addresses your point!

---

> > > > ### Author Response · Authors · 2022-08-02
> > > > **Response to Reviewer sqR1 (4 of 5)**
> > > >
> > > > > “5. The practical UNet has different number of channels in different layers. Does this work also support the different number of channels?”
> > > >
> > > > Yes, our work generalises to multiple latent channels, which are commonly used in practical U-Nets with, say, convolutional neural networks as parameterisations, and is mathematically justified in this setting.
> > > > We do this in an analogous way to how analysis for grayscale image approximations extend to multiple colour channels, which is mentioned in footnote 2 on page 3.
> > > > Furthermore, our framework applies to VDVAE, NVAE and Markovian HVAEs (see for instance Appendix A.5, Fig. A.1) which all use convolutional neural networks with varying latent channels as the parameterising functions.
> > > > We now refer precisely to this content—thanks to your other remark—in the main text in l. 184 and l. 196.
> > > > Thank you for your thoughtful question which we believe helps improve the exposition of our framework!
> > > >
> > > > - - -
> > > >
> > > > > “6. An important feature of UNet is the skip connection across different layers. It seems that the framework in this paper ignores this feature.”
> > > >
> > > > Thank you for raising this important question on whether our framework takes into account skip connections.
> > > > Our framework covers and (in the absence of an imposed invertibility constraint) even requires skip connections as we will explain below.
> > > > However, we fully agree that we should have communicated this better in our submission, and have hence updated our manuscript to reflect this (see Appendix B.2, and particularly Remark 2).
> > > >
> > > > In Appendix B.2, as the working definition of a U-Net, we use $f_{\theta,j}$ and $b_{\theta,j}$ to refer to the forward (encoder) and backward (decoder) on resolution/latent space $V_{-j}$.
> > > > One goal of the decoder $b_{\theta,j}$ is to reconstruct the data from $f_{\theta,j}$, or, put differently, the decoder shall “invert” the encoder.
> > > > This can be achieved in two ways:
> > > > First, it can be imposed directly, such as in Normalizing flows.
> > > > Second, it can be approximated via skip connections.
> > > > Intuitively, the encoder passes the decoder a ‘target point’ of where to move in space through these skip connections.
> > > > The decoder is then constrained to follow the encoder's path, but in the inverse direction.
> > > > In conclusion, our U-Net framework covers and requires skip connections in this second case to ensure that a (learnt) backward process inverts a (learnt) forward process.
> > > >
> > > > Thank you for this feedback which very much helped improve our manuscript. We hope this explanation and the changes we have made resolve your concern.
> > > >
> > > > - - -
> > > >
> > > > > “7. This paper relies on heavy notations. These notations should be explained more. For example, the author could explain what the practical counterpart is when introducing a notation.”
> > > > > “The paper builds upon heavy notations with less explanation. Therefore, I think the writing quality and clarity can be improved.”
> > > >
> > > > Thank you for this valuable feedback!
> > > > We agree that our work is somewhat notation heavy, and propose three ways to accommodate this:
> > > > First, we would like to refer to Appendix A.1 in our submission on “Definitions and Notations”, which we hope is useful towards exactly this point.
> > > > In general, Appendix A provides an in-depth introduction to our framework, including technical proofs. Each Theorem in the main text is often preluded by a series of Lemmas and Definitions which motivate it, and we hope this Appendix is useful towards understanding our notation in detail.
> > > > Second, the new Appendix B.2 aims to better explain the practical counterparts of our notation in a U-Net (referring to point 4. for details).
> > > > Third, we have made several changes in our manuscript to simplify and improve the presentation of our theoretical results.
> > > > In particular, we have written the new Appendix B, particularly referring to Appendix B.2 for a detailed introduction to our notation used on U-Nets, simplified the notation throughout Appendix A where possible, and amended Theorem 3 to use the in machine learning much more common concepts of the Wasserstein-2 metric and Lipschitz continuity.
> > > >
> > > > We hope these changes contribute towards the readability of our work, and again thank you for this feedback!

---

> > > > > ### Author Response · Authors · 2022-08-02
> > > > > **Response to Reviewer sqR1 (5 of 5)**
> > > > >
> > > > > > “8. When the author refers to the Appendix, the author could refer to a more detailed position.”
> > > > >
> > > > > We agree with your observation.
> > > > > We thoroughly addressed this point, and kindly refer you to our “Response to all Reviewers” (Section “Similar question by multiple reviewers”) above for the details.
> > > > >
> > > > > - - -
> > > > >
> > > > > > “9. This paper relies on UNet, HVAE and multi-resolution partition. It would be better to add a background section to introduce these important prior works. This would make this paper more readable.”
> > > > >
> > > > > We thank you for this great suggestion. We agree that a background section would be beneficial, also in response to your other comment on U-Nets. We initially considered adding such a background section, but were then worried our paper would become too long. Instead, we had (partially) provided such background in Appendices of our submission: on multi-resolution partition (jointly with an explanation and illustration on the thought experiment) in (old=submission) Appendix B, and on HVAEs in (old) Appendix C.
> > > > >
> > > > > We follow your feedback, and have dedicated a new Appendix (new=revision Appendix B) entirely to Background, which consists of three parts:
> > > > >
> > > > > * Multi-resolution Partition (Appendix B.1), resembling the content of (old) Appendix B.
> > > > > * U-Nets (Appendix B.2), which is an entirely new section, explaining our notation (e.g. of forward and backward transition kernels $F_j$ and $B_j$), and their practical complements in the U-Net literature (e.g. parameterised neural networks such as convolutional neural networks), and a definition of a U-Net (see Definition 4).
> > > > > * HVAEs (Appendix B.3), resembling the content of (old) Appendix C.
> > > > >
> > > > >
> > > > > In a potential camera-ready version of this manuscript, we will consider adding (a condensed version) of this Appendix in a dedicated Background section in the main text, using the extra page granted. – Thank you again for this great idea which we hope improves the understanding of the necessary literature for the reader of our paper.

---

> > > > ### Comment · Reviewer_sqR1 · 2022-08-07
> > > > **On Q3**
> > > >
> > > > Thanks for the reply. It seems there is still some ambuiguity. If I use $V_j'$ to refer to the multi-resolution partition of $L^2(X)$ and use $V_j$ to refer to the multi-resolution partition of $L^2(R^m)$, which one is correct: $V_j' = \\{f \in V_j: supp (f) = X  \\}$ or $V_j' = \\{f \cdot 1_X: f \in V_j  \\}$?

---

> > > > > ### Author Response · Authors · 2022-08-08
> > > > > **Author response to “On Q3”**
> > > > >
> > > > > When we define a multi-resolution hierarchy (before: partition) over $\mathbb{X}$, like for any linear subspace, you may use the basis restricted to $\mathbb{X}$, that is any basis function outside of $L2(\mathbb{X})$ is zero (the first version of $V_{j}^{’}$ in your comment). Alternatively, if you prefer, you can still use the multi-resolution hierarchy over $\mathbb{R}^n$ and just consider the approximation of a compactly supported function whilst using the original basis (the second version of $V_{j}^{’}$ in your comment). In both cases, the approximant over the compact set is the same. Hence, there is no ambiguity as the choice is inconsequential.

---

> > > > ### Comment · Reviewer_sqR1 · 2022-08-07
> > > > **On Q4**
> > > >
> > > > I have read Appendix B.2. In Eq.(B.106), the idealized UNet has a chain structure. It seems that the chain structure can't represent the skip connection. Also, it is not supported how the assumption $B_j F_j = I$ can be approximated by skip connection.
> > > >
> > > > I didn't find new Remark 2.

---

> > > > > ### Author Response · Authors · 2022-08-08
> > > > > **Author response to “On Q4”**
> > > > >
> > > > > > “I didn't find new Remark 2.”
> > > > >
> > > > > First of all, apologies: We meant Remark 1 instead of Remark 2 in B.2 right below Definition 4 of the idealised U-Net.
> > > > >
> > > > > > “I have read Appendix B.2. In Eq.(B.106), the idealized UNet has a chain structure. It seems that the chain structure can't represent the skip connection.”
> > > > >
> > > > > While Eq. (B.106) is a composition of functions, seemingly a “chain structure [without skip connections]”, implicit to it are skip connections between the forward and the (first) backward pass on the same resolution. More precisely, $B_{j,\theta}$ “receives as input the output of” $F_{j,\theta}$ on the same resolution $j$. The skip connections are also illustrated in Fig. B.3 which we hope helps understanding this point. In summary, the U-Net which we theoretically analyse in our framework considers skip connections.
> > > > >
> > > > > > “Also, it is not supported how the assumption $B_j F_j = I$ can be approximated by skip connection.”
> > > > >
> > > > > Thank you for this question. At one stage we did have a paragraph in our main text which elaborated on this point, but in the end we removed it to save space. Below, we provide a revised, shortened version of it to attempt to clarify this point:
> > > > >
> > > > > Consider a U-Net without a dimension reduction and embedding operation.
> > > > > In this case, the problem is underdetermined, and $B_j F_j = I$ can even be achieved trivially, i.e. independent of the data, yielding perfect reconstructions.
> > > > > Now assume we introduce one step of dimension reduction and embedding, hence now have a very simple (perhaps the simplest) U-Net and a bottleneck.
> > > > > Here, the reconstruction problem can be overdetermined.
> > > > > To solve this issue and make the problem underdetermined again, U-Nets introduce skip connections.
> > > > >
> > > > > The way U-Nets are used for unconditional sampling, such as in HVAEs, is by finding some way to “forget” the reliance on the skip connections while still producing high-quality samples.
> > > > > This is done through the second backward pass in Appendix B.2, corresponding to the “prior distribution” in HVAEs, which is enforced to be close to the first backward pass, the “approximate posterior”, via a KL-divergence.
> > > > > Hence, when identifying the regularision property, the assumption $B_j F_j = I$ just formalises the premise that with skip connections, we get good reconstructions (which in practice we usually do very easily), and now ask what happens when the skip connections are absent, as is the case in the second backward pass during unconditional sampling.
> > > > > Importantly, the assumption $B_j F_j = I$ pertains to each approximation space $V_j$, as this is where skip connections bridge.
> > > > >
> > > > > It is also worth noting that we are working on an extension of our theory on the regularisation property to cases where, for instance, $B_j F_j = I$, is violated. However, we believe that for now, $B_j F_j = I$ is a reasonable and intuitive assumption. Furthermore, it simplifies the technical concepts in the paper to highlight the general ideas present in the U-Net structure in a more approachable way, without the confusion of looking at more complicated cases in the already long Appendix of the work. As it stands, this work is a first step towards analysing U-Nets, and we hope work by many others will be sparked by it who will extend our framework and the underlying assumptions in various ways.
> > > > >
> > > > > In a potential camera-ready version of this paper, we will include a paragraph explaining the above in detail. We hope this resolves your concern!

---

> > > ### Comment · Reviewer_sqR1 · 2022-08-07
> > > **On Q2**
> > >
> > > Thanks for the detailed explanation, which addressed my misunderstanding in Q2.
> > >
> > > Definition 1 is not easy to understand for those less familiar image processing, but I think Eq.(B.104) is a very intuitive example. So I suggest  move Eq.(B.104) to the main paper, which helps understand Definition 1 and L87-96.

---

> > > > ### Author Response · Authors · 2022-08-08
> > > > **Author response to “On Q2”**
> > > >
> > > > Great, we appreciate the confusion on ‘zooming in’ could be resolved!
> > > >
> > > > We very much like your suggestion of including Eq. (B.104) into the main text for two reasons: First, we fully agree that it helps understanding Definition 1. Second, it precisely formalises the revised “intuitive explanation” of “piecewise constant functions” in “Author response to Q1”, and hence likewise contributes to a better understanding of the points discussed there. Hence, we have included it in ll. 91. Thank you again!

---

> > ### Comment · Reviewer_sqR1 · 2022-08-07
> > **On Q1**
> >
> > Thanks for the detailed explanation, and I have understand the orthogonal to a finite dimensional function space. However, I still have some confusion.
> >
> > In L64, the author mentioned the pixel-space projection, i.e., mapping an image to finite resolutions $T(f) = \\{ f(i, j) \\}_{i,j}$, where $i,j$ come from a finite space of $R^2$.
> >
> > The multi-resolution projection is done in the function space, i.e., $T(f) = \sum_{k=1}^K a_k g_k$, where $\{g_k\}_{k=1}^\infty$ is a basis of $L^2$.
> >
> > From L60-71, it seems that the author means the former pixel-space projection can be achieved by the functional-space projection. Nevertheless, I have no idea how this can be achieved.

---

> > > ### Author Response · Authors · 2022-08-08
> > > **Author response to “On Q1”**
> > >
> > > Thank you for engaging with us!
> > >
> > > We are glad and thank you for having worked with us to resolve the confusion around finite-dimensional projections, and are glad this point is resolved!
> > >
> > > Furthermore, thanks to your last message, we believe we now fully understand where the confusion on point 1. is rightly resulting from, and it is exactly the sentences in ll. 60 which you are referring to. For ease of reference, we cite them here: “A grayscale image with infinite resolution can be thought of as the graph of a two-dimensional function over the unit square. To store these infinitely-detailed images in computers, we project them to some finite resolution, e.g. $512^2$ pixels. *These projections can still be thought of as the graphs of functions, but now a discrete one and only defined at a finite number of grid points*. The relationship between the discretised version and its infinitely-fine counterpart depends entirely on how we construct this projection to preserve the details we wish to keep. One approach is to prioritise preserving the large-scale details of our images, so unless closely inspected, the projection is indistinguishable from the original. This can be achieved with a multi-resolution projection of the image [l. 60-68].”
> > >
> > > While we are not sure, we believe the sentence highlighted in *italic* is most definitely the reason why you suggested the functions in our multi-resolution framework to have domain $\mathbb{Z}^2$ rather than $\mathbb{R}^2$. When just reading this sentence, it is indeed confusing, we fully agree. The functions $f \in V_j$ are not “defined” over a “grid”, i.e. their domain is not $\mathbb{Z}^2$ as this sentence may imply. Instead, what we wanted to illustrate is that these functions consist of finitely many piecewise constant intervals over the domain $\mathbb{R}^2$ (i.e. they are piecewise-constant functions). When storing these mathematical objects in a computer, we store the function values taken at these finitely many intervals, the ‘pixels’, as a “grid”, i.e. the usual array object we practically work with when processing images in deep learning. Further, as there seems to be a concern whether the above is at all possible or reasonable, we note that proceeding like this is standard in the context of Haar wavelets and image processing.
> > >
> > > More technically speaking, functions in the approximation spaces $V_j$ are piecewise constant on a dyadic square, i.e. a set of the form $[2^j(i),2^j(i+1)] \times [2^j(k),2^j(k+1)]$ for $i,k = 0,…,2^{(-j)-1}$ (over the unit square). When the image is supported on $[0,1]^2$, there are $2^{-j} \cdot 2^{-j}$ many coefficients, and we put these into an array in our computer. If this is taken to be a matrix, it is a bijection. – This is precisely how the “pixel-space projection can be achieved by the functional-space projection [on Q1].”
> > >
> > > To reflect the above explanation and avoid misunderstandings, we have made the following change to the paragraph at hand in our latest revision: “A grayscale image with infinite resolution can be thought of as the graph of a two-dimensional function over the unit square. To store these infinitely-detailed images in computers, we project them to some finite resolution. These projections can still be thought of as the graphs of functions with support over the unit square, but they are piecewise constant on finitely many intervals or ‘pixels’, e.g. $512^2$ pixels, and we store the function values obtained at these pixels in an array or `grid’. The relationship between the finite-dimensional version and its infinitely-fine counterpart depends entirely on how we construct this projection to preserve the details we wish to keep. One approach is to prioritise preserving the large-scale details of our images, so unless closely inspected, the projection is indistinguishable from the original. This can be achieved with a multi-resolution projection of the image.”
> > >
> > > We hope this resolves the confusion, also in the manuscript, and we thank you for pointing this out and helping us improve the work!

---

> ### Author Response · Authors · 2022-08-08
> **Any remarks on our responses to 5. to 9.?**
>
> Dear reviewer sqR1,
>
> Thank you again for engaging with us! We just wanted to check whether our author responses and amendments on 5. to 9. address your questions and concerns?
>
> Many thanks!

---

> > ### Comment · Reviewer_sqR1 · 2022-08-08
> > **Thanks for the reply.**
> >
> > Thanks for the detailed reply. Can you give an instantiation of Eq.(B.106), i.e., specifying $F_j, B_j, P_j, E_j$, that explicitly includes the skip connection in the expression of U?
> >
> > I have no concerns on other questions.

---

> > > ### Author Response · Authors · 2022-08-09
> > > **Author response to “Thanks for the reply.”**
> > >
> > > > “Can you give an instantiation of Eq.(B.106), i.e., specifying $F_j, B_j, P_j, E_j$, that explicitly includes the skip connection in the expression of U?”
> > >
> > > We are happy to provide such an instantiation.
> > > Consider the following scenario:
> > > Assume the data lives in $\mathbb{R}^2$ and we use the notation $(x,y)$ for a datum.
> > > We now consider a simplified U-Net with just one dimensionality reduction step $P$ and corresponding embedding step $E$, and a forward process $f$ and backward process $b$ on a single resolution.
> > > Assume $f(x,y) = y/2, P(x,y)=x, E(x) = (x,0), b(x,y) = 2y$.
> > > Let $F$ and $B$ be the push forwards of $f$ and $b$.
> > > Now the composition $U = B E P F \neq I$ as $A = \\{ (x,y) | y = 0 \\}$ is invariant under $B$.
> > > However, in the presence of skip connections, we can invert such an F as the skip passes information not on $A$, that is $BF = I$.
> > > Intuitively speaking, $B$ can’t ‘move’ anything in the set of $A$ outside of the set of $A$.
> > > This means that at training time, we can produce good reconstructions when using the skip connection.
> > > The goal of our theoretical analysis in Theorem 3 is, assuming good reconstructions are ubiquitous in practice (i.e. assuming $B_j F_j = I$), what happens when we need to forget the dependence of the skip connection.
> > > We expect a trade-off between reconstructions and unconditional sampling, where in current instantiation of U-Nets good reconstructions are common and unconditional samples can be poor.
> > > We have formalised this observation with the assumption $B_j F_j = I$, but in future work, we intend to precisely quantify this trade-off.
> > >
> > > We will consider providing this example and perhaps more illustration on the definition of a U-Net in a potential camera-ready version. Thank you for your great question and the discussion!
> > >
> > >
> > > > “I have no concerns on other questions.”
> > >
> > > We are very pleased that all other concerns could be resolved, and thank you for helping us to improve and clarify the submission during the process!
> > >
> > > We would be glad if you would consider this in your final evaluation of our work, and consider increasing your overall score.

---

### Official Review · Reviewer_fuoh · 2022-07-12

**Rating:** 8
**Confidence:** 3
**Soundness:** 4 excellent
**Presentation:** 4 excellent
**Contribution:** 4 excellent

**Summary:**

The paper proposes a multi-resolution framework for U-Net architecture, which provides some insights into the architecture: (1) it can relate UNets as diffusion processes (2) average pooling regularizes it by implicitly learning Haar wavelet basis representation of target data. The authors analyze Hierarchical Gaussian VAE (HVAE) by applying the proposed framework to the model, which can explain the instability of sampling in HVAE and helps reduce parameter redundancies.


**Questions:**

- Sampling instabilities on multi-modal datasets such as CIFAR10 and ImageNet are really due to the instability from the hierarchical property? VAE w/o UNet structure also usually suffers from poor sampling performance on such vision datasets.

**Limitations:**

Limitations are addressed, no negative societal impact observed.

**Strengths And Weaknesses:**

### Strong points
Although why UNet has been successful in many applications, its regularization effect is poorly understood ever. The framework can naturally explain the instability issue and parameter redundancies of HVAE. The proposed framework is not only elegant but also insightful. In addition, experimental results are overall convincing and support the theory part.
The presentation of the results is correct and easy to follow.

### Weak points
- My only issue is the connection between the insight into sampling instabilities and empirical observation in Sect. 3.3 is unclear (please see my question).
- It is not easy to refer to corresponding appendix parts (because appendix. A is long and the numbering for subsection is not used).

---

> ### Author Response · Authors · 2022-08-02
> **Response to Reviewer fuoh**
>
> We are glad you like our work! We thank you for highlighting our framework as elegant and insightful and are glad you found our work easy to follow. We would like to address the questions stated in your review below, noting that some have been addressed in the “Response to all reviewers” above.
>
> - - -
>
> > “My only issue is the connection between the insight into sampling instabilities and empirical observation in Sect. 3.3 is unclear [...] Sampling instabilities on multi-modal datasets such as CIFAR10 and ImageNet are really due to the instability from the hierarchical property? VAE w/o UNet structure also usually suffers from poor sampling performance on such vision datasets.”
>
> Thank you for this question. We should have been clearer about the exact root cause of the sampling instabilities observed - we have revised the paper to reflect this (ll. 223) by adding content based on the explanation provided below.
>
> We completely agree that this issue is not exclusive to HVAEs or U-Nets, rather it is a consequence of flowing a data distribution to one with different topological properties (see for instance [6] for a similar argument on Normalizing flows). In current HVAEs, one attempts to “squish” the highly complex distribution of images to a point mass at zero (see Theorem 5), which is perhaps an extreme case of the instability studied in [6]. With this perspective in mind, it is intuitive that similarly constructed single-layer VAEs without a hierarchical latent variable structure underperform on these datasets.
>
> Yet, one might then think that the rich parametric form of HVAEs would solve this problem – after all, the prior and the variational posterior are both autoregressive over latent layers, which might perhaps seem to be a route out from these instabilities. We wanted to express that this problem is still present for HVAEs such as VDVAE as they connect their data distribution to a point mass, which is not explicitly expressed in prior work. This insight is identified by the theoretical analysis of HVAEs our framework provides.
>
> - - -
>
> > “It is not easy to refer to corresponding appendix parts (because appendix. A is long and the numbering for subsection is not used).”
>
> Thank you for raising this valuable point. We have now thoroughly addressed this point in our latest revision (kindly see “Response to all reviewers”, section “Similar question by multiple reviewers”).

---

### Author Response · Authors · 2022-08-02
**Response to all reviewers (1 of 2)**

### Introduction

We thank the reviewers for their encouraging and valuable feedback. The reviews have motivated us to revise and improve our manuscript. We respond to each review separately below and have revised the manuscript to incorporate your feedback.

We first provide a short summary to reinforce the main points and contributions of our work. This is relevant to all reviewers and the area chair:

Our work provides theory for understanding the empirically successful and widely used U-Net architecture. U-Nets are a core building block in state-of-the-art generative models such as hierarchical VAEs (HVAEs) and score-based diffusion models [3,4,7,8,9], but prior to our work there was a lack of analysis and theoretical insight into why our field makes this choice. This work provides the first theoretical framework for this analysis, and characterises important observations towards this goal. More specifically, we

A.  identify the regularisation property of U-Nets with average pooling as learning a representation over a Haar wavelet basis,
B.  identify HVAEs (VDVAE [4], NVAE [3], Markovian HVAEs [1,2]) with a U-Net architecture as forward Euler discretisations of a coupled diffusion process flowing to a point mass, drawing an important connection to seemingly unrelated model classes in probabilistic deep learning, for example score-based diffusion models, which all—as we find—approximate a diffusion process in different ways; and state hypotheses on sampling instabilities and parameter redundancy of these models, and we
C.  empirically validate these hypotheses in our experiments and ablation studies, and further show how HVAEs ‘secretly’ learn a representation of time in their residual state which they make use of during training.

We therefore provide a first step towards an in-depth understanding of U-Nets and their theoretical properties which we hope will spark and enable further analysis and understanding on this topic, in turn facilitating the additional potential for empirical performance gains.

### Overview of key changes in our revision of the submission

* new Appendix B on 1) U-Nets, our formalisation of them and their practical counterparts (including an illustration of U-Nets and our notation in Figure B.3, a Definition of U-Nets in Definition 4, and a remark on skip connections in Remark 2), 2) HVAEs, and 3) Multi-resolution Partition
* simplified the notation and terminology where possible in various locations of the manuscript, particularly in Appendix A, for instance replacing “multi-resolution partition” with “multi-resolution hierarchy”, which hope aids the understanding of our thought experiment
* amended Definition 1 to define $L^2(\mathbb{X})$ via the inclusion property (in response to Reviewer sqR1, point 3.)
* simplified Theorem 3, one of our key results, to now use the in machine learning much more commonly used Wasserstein-2 metric and Lipschitz continuity

---

> ### Author Response · Authors · 2022-08-02
> **Response to all reviewers (2 of 2)**
>
> ### Similar question by multiple reviewers
>
> > Reviewer fuoh: “It is not easy to refer to corresponding appendix parts (because appendix. A is long and the numbering for subsection is not used).”
>
> > Reviewer sqR1: “8. When the author refers to the Appendix, the author could refer to a more detailed position.”
>
> We thank you for raising this and agree that the submitted manuscript would benefit from more exact Appendix references. We have hence thoroughly addressed it in the latest revision of our paper. In general, we amended all existing References to Appendix A to point to specific Subsections/Equations/Figures to facilitate a better interconnection between the main text and the Appendix.
>
> In particular, we have made the following changes to existing references of “Appendix A”:
>
> * ll. 111, on the definition of weak continuity used; now: “see Appendix A.1, Eq. (A.5)”
> * ll. 122, on the notation of a diffusion transition kernel; deleted.
> * ll. 128, on Theorem 1; now: “[...] (see Appendix A.4 [...].”
> * l. 140, on the standard basis; now: “[...] (see Appendix A.2, Eq. (A.9)).”
> * ll. 143, on the Haar basis; now: “[...] see Appendix A.2.”
> * l. 147, on Theorem 2; now: “[...] in Appendix A.2”
> * ll. 184, on the generalisation of our Framework and our theoretical results (particularly Theorem 4)  to Markovian HVAEs [1,2] and NVAE [3], the former referring to the perhaps simplest HVAE design, which however performs poorly in practice, and the latter referring to another state-of-the-art HVAE model; now: “[...] in Appendix A.5.”
> * l. 196, on the discretisation scheme of VDVAE, NVAE and Markovian HVAEs; now: “[...] in Appendix A.5, Fig. A.1.”
> * l. 229, on the bolstering of the rate of convergence of $Y_i \xrightarrow{t \to 1} \infty$ through the term $Z_{i,+}^{(\sigma)}$ which is present in VDVAE, but absent in NVAE; deleted, instead referring to Fig. A.1 in the second sentence thereafter
> * l. 236, on the time-homogeneous model formulation; now: “see Appendix (A.7)”.
> * ll. 558, checklist item on assumptions of all theoretical results; added a reference to the new Appendix B on Background in response to reviewer sqR1.
>
> We thank you for this valuable remark which ensures a better interconnection between Appendix A and the main text.
>
> ### References
>
> All references to lines (l. … or ll. …) in our responses refer to the revised manuscript, if not stated otherwise.
>
> Throughout our response, we use the following literature references:
>
> [1] Casper Kaae Sønderby, Tapani Raiko, Lars Maaløe, Søren Kaae Sønderby, and Ole Winther. Ladder variational autoencoders. In D. Lee, M. Sugiyama, U. Luxburg, I. Guyon, and R. Garnett, editors, Advances in Neural Information Processing Systems, volume 29. Curran Associates, Inc., 2016.
>
> [2] Yuri Burda, Roger Grosse, and Ruslan Salakhutdinov. Importance weighted autoencoders. arXiv preprint arXiv:1509.00519, 2015.
>
> [3] Arash Vahdat and Jan Kautz. NVAE: A deep hierarchical variational autoencoder. In Advances in Neural Information Processing Systems, volume 33, 2020.
>
> [4] Rewon Child. Very Deep VAEs Generalize Autoregressive Models and Can Outperform Them on Images. In International Conference on Learning Representations, 2021.
>
> [5] Ingrid Daubechies. Ten lectures on wavelets. SIAM, 1992.
>
> [6] Cornish, R., Caterini, A., Deligiannidis, G. and Doucet, A., 2020, November. Relaxing bijectivity constraints with continuously indexed normalising flows. In International conference on machine learning (pp. 2133-2143). PMLR.
>
> [7] Jonathan Ho, Ajay Jain, and Pieter Abbeel. Denoising diffusion probabilistic models. In Advances in Neural Information Processing Systems, volume 33, 2020.
>
> [8] Yang Song, Jascha Sohl-Dickstein, Diederik P Kingma, Abhishek Kumar, Stefano Ermon, and Ben Poole. Score-based generative modeling through stochastic differential equations. arXiv preprint arXiv:2011.13456, 2020.
>
> [9] Valentin De Bortoli, James Thornton, Jeremy Heng, and Arnaud Doucet. Diffusion schrödinger bridge with applications to score-based generative modeling. In Advances in Neural Information Processing Systems, volume 34, 2021

---

### Author Response · Authors · 2022-08-06
**Author-reviewer discussion - We would very much appreciate discussing our author responses!**

Dear reviewers,

Once again, thank you very much for your valuable time spent on our submission and your thoughtful reviews!

As the Author-Reviewer discussion period is coming to an end soon, we wanted to check in with you if we have addressed your questions and concerns, and if we provided all information required for making your final evaluation.
In particular, we believe our response to reviewer sqR1 requires the attention of the reviewers as the mentioned concerns constituting the low overall, initial score (borderline reject) are either clarified or addressed, and should be considered towards the final decision.
We would appreciate interacting with you on these and other points from the reviews and our author responses.

Thank you again for your efforts!

---

### Meta-Review · Area_Chair_Ae62 · 2022-08-29

**Recommendation:** Accept
**Confidence:** Less certain

**Metareview:**

The paper makes an observation that average pooling in U-Nets implicitly learn a Haar wavelet basis representation and build a theory for hierarchical VAEs (HVAEs) on top of it. The proposed interpretation of HVAEs lead to modification to HVAEs that reduce the number of parameters and improve stability. I think the Haar wavelet basis representation is somewhat obvious but the analysis of HVAEs look nontrivial. I would recommend accepting this paper with the following suggestions:

I suggest the authors to focus on the HVAE part and improve the presentation. More specifically, the definition of U-Net architectures used in HVAEs should be in the main text. Also make more connection between HVAEs and diffusion models. For example, time information is explicitly handled in diffusion models, whereas this paper suggest that it is handled implicitly in HVAEs. The parameter sharing is common for diffusion models. At the end of the day, the proposed improvement to HVAEs seem to make HVAEs closer to diffusion models.


**Award:**

No

---

### Decision · Program_Chairs · 2022-09-14

Accept